# Graph Neural Networks with Adaptive Readouts

**David Buterez** [*1]  **Jon Paul Janet** [2]  **Steven J. Kiddle** [3]  **Dino Oglic** [3]  **Pietro Liò** [1]

[1] Department of Computer Science and Technology, University of Cambridge, UK
[2] CVRM, BioPharmaceuticals R&D, AstraZeneca, Sweden
[3] DS&AI, BioPharmaceuticals R&D, AstraZeneca, UK

## Abstract

An effective aggregation of node features into a graph-level representation via readout functions is an essential step in numerous learning tasks involving graph neural networks. Typically, readouts are simple and non-adaptive functions designed such that the resulting hypothesis space is permutation invariant. Prior work on deep sets indicates that such readouts might require complex node embeddings that can be difficult to learn via standard neighborhood aggregation schemes. Motivated by this, we investigate the potential of adaptive readouts given by neural networks that do not necessarily give rise to permutation invariant hypothesis spaces. We argue that in some problems such as binding affinity prediction where molecules are typically presented in a canonical form it might be possible to relax the constraints on permutation invariance of the hypothesis space and learn a more effective model of the affinity by employing an adaptive readout function. Our empirical results demonstrate the effectiveness of neural readouts on more than 40 datasets spanning different domains and graph characteristics. Moreover, we observe a consistent improvement over standard readouts (i.e., sum, max, and mean) relative to the number of neighborhood aggregation iterations and different convolutional operators.

## 1 Introduction

We investigate empirically the potential of adaptive and differentiable readout functions for learning an effective representation of graph structured data (e.g., molecular, social, biological, and other relational data) using graph neural networks (GNNs). Recently, there has been a surge of interest in developing neural architectures from this class [1–5]. Graph neural networks typically employ a permutation invariant neighborhood aggregation scheme that is repeated for several iterations, where in each iteration node representations are updated by aggregating the feature vectors corresponding to their neighbors. The process is repeated for a pre-specified number of iterations and the resulting node representations capture the information contained in the respective neighborhoods given by vertex rooted sub-trees. The final step is aggregation of node features into a graph-level representation using a readout function. The readouts are typically selected such that the resulting hypothesis space is permutation invariant. For instance, simple functions such as sum, mean, and max, all satisfy this requirement [6, 7]. Graph neural networks can, thus, be seen as a special case of representation learning over (node) sets. Zaheer et al. [6] have studied learning on sets and demonstrated that a permutation invariant hypothesis over such domains admits a decomposition as a sum of individual set items represented in a latent space given by a suitable embedding function. In a follow up work, Wagstaff et al. [7] have demonstrated that simple pooling/readout functions such as sum, mean, or max might require complex item/node embedding functions that might be difficult to learn using standard neural networks. The expressiveness of graph neural networks specifically has also been studied in [5], where it has been recommended to use an injective neighborhood aggregation scheme.

---

[*]Correspondence to: David Buterez <db804@cam.ac.uk>.

36th Conference on Neural Information Processing Systems (NeurIPS 2022).

For such schemes, it can be demonstrated that graph neural networks can be as expressive as the Weisfeiler–Lehman isomorphism test which is known to be an effective and computationally efficient approximation scheme for differentiating between a large number of graph isomorphism classes [8, 9].

As it can be challenging to learn a permutation invariant hypothesis over graphs using simple readouts, we empirically investigate possible extensions and relaxations for problems where graphs might be presented in a canonical form (i.e., with an identical ordering of vertices). In such cases, it might be possible to relax the constraint on permutation invariance of the hypothesis space. For instance, in problems such as binding affinity prediction, molecular graphs are typically generated from a canonical SMILES representation and, thus, inputs to graph neural networks are graphs with a fixed ordering of nodes. The latter is sufficient to ensure consistent predictions over molecular graphs for graph neural networks with readouts that do not give rise to permutation invariant hypotheses.

We start with a review of graph neural networks and then focus on introducing different classes of adaptive and differentiable readout functions. The first class of such readouts is based on set transformers [10] and it gives rise to permutation invariant hypotheses. We then slightly relax the constraint on permutation invariance of graph-level representations by introducing readouts inspired by a neighborhood aggregation scheme known as Janossy pooling [11]. These approximately permutation invariant readouts are based on multi-layer perceptrons (MLP) and recurrent neural architectures known as GRUs. Finally, we consider neural readouts based on plain MLP and GRU architectures, thus completely lifting the constraint on permutation invariance of the hypothesis space.

Our empirical study is extensive and covers more than 40 datasets across different domains and graph characteristics. The ultimate goal of the study is to explore the potential of learning hypotheses over graph structured data via adaptive and differentiable readouts. To this end, we first consider the most frequently used neighborhood aggregation schemes or convolutional operators and fix the number of iterations to two. Our empirical results demonstrate a significant improvement as a result of employing neural readouts, irrespective of the convolutional operator and dataset/domain. Following this, we then compare our neural readouts to the standard readouts (sum, max, mean) while varying the number of neighborhood aggregation iterations. The results indicate that neural readout functions are again more effective than the standard readouts, with a significant difference in performance between different neighborhood aggregation schemes. We hypothesize that the latter might be due to the expressiveness of different neighborhood aggregation operators. More specifically, in drug design and lead optimization it is typical that through a change in sub-structure of a parent compound that one can improve the potency. These changes are local and we hypothesize that they would be reflected in a small number of node representations, whose signal could be consumed by a large number of noisy nodes within the standard readouts. We also present results of an ablation study on the influence of node permutations on the hypotheses learned by GNNs with neural readouts that do not enforce permutation invariance. The results indicate that there might be some node permutations that are detrimental to the predictions and this might be an interesting avenue for future work.

We conclude with an experiment involving multi-million scale proprietary datasets from AstraZeneca that have been collected by primary screening assays. Our results again demonstrate that the avenue of neural readouts merits further exploration from both theoretical and empirical perspectives. More specifically, we observe that only plain MLP readouts significantly improve the performance on these challenging tasks and they do not give rise to permutation invariant hypotheses. Extensive results, including additional datasets and neural architectures (variational graph autoencoders and visualizations of latent spaces that correspond to different readouts) have been provided in Appendix L. An analysis involving computational and memory costs and trade-offs can be found in Appendix I.

## 2 Graph Neural Networks with Adaptive and Differentiable Readouts

Let $\mathcal{G} = (\mathcal{V}, \mathcal{E})$ be a graph, where $\mathcal{V}$ is the set of *nodes* or *vertices* and $\mathcal{E} \subseteq \mathcal{V} \times \mathcal{V}$ is the set of *edges*. Suppose the nodes are associated with $d$-dimensional feature vectors $\mathbf{x}_u$ for all $u \in \mathcal{V}$. Let $A$ be the adjacency matrix of a graph $G$ such that $A_{uv} = 1$ if $(u, v) \in \mathcal{E}$ and $A_{uv} = 0$ otherwise. For a vertex $u \in \mathcal{V}$ denote the set of neighboring nodes with $\mathcal{N}_u = \{v \mid (u, v) \in \mathcal{E} \vee (v, u) \in \mathcal{E}\}$. Suppose also that a set of graphs with corresponding labels $\{(G_i, y_i)\}_{i=1}^n$ has been sampled independently from some target probability measure defined over $\mathcal{G} \times \mathcal{Y}$, where $\mathcal{G}$ is a space of graphs and $\mathcal{Y} \subset \mathbb{R}$ is the set of labels. We are interested in the problem of learning a graph neural network that can approximate well the target label $y \in \mathcal{Y}$ for a given graph $G \in \mathcal{G}$.

Henceforth, we will assume that a graph $G$ is represented with a tuple $(X_G, A_G)$, with $X_G$ denoting the matrix with node features as rows and $A_G$ the adjacency matrix. Graph neural networks take such tuples as inputs and generate predictions over the label space. A function $f$ defined over a graph $G$ is called permutation invariant if there exists a permutation matrix $P$ such that $f(PX_G, PA_GP^\top) = f(X_G, A_G)$. In general, graph neural networks aim at learning permutation invariant hypotheses to have consistent predictions for the same graph when presented with permuted vertices/nodes. This property is achieved through neighborhood aggregation schemes and readouts that give rise to permutation invariant hypotheses. More specifically, the node features $X_G$ and the graph structure (adjacency matrix) $A_G$ are used to first learn representations of nodes $h_v$, for all $v \in \mathcal{V}$. The neighborhood aggregation schemes enforce permutation invariance by employing standard pooling functions — sum, mean, or max. This step is followed by a readout function that aggregates the node features $h_v$ into a graph representation $h_G$. As succinctly described in [5], typical neighborhood aggregation schemes characteristic of graph neural networks can be described by two steps:

$$a_v^{(k)} = \text{AGGREGATE}(\{h_u^{(k-1)} \mid u \in \mathcal{N}_v\}) \quad \text{and} \quad h_v^{(k)} = \text{COMBINE}(h_v^{(k-1)}, a_v^{(k-1)}) \tag{1}$$

where $h_u^{(k)}$ is a representation of node $u \in \mathcal{V}$ at the output of the $k^{\text{th}}$ iteration. For example, in graph convolutional networks the two steps are realized via mean pooling and a linear transformation [1]:

$$h_v^{(k)} = \sigma \left( \frac{1}{|\mathcal{N}_v^*|} \sum_{u \in \mathcal{N}_v^*} W^{(k)} h_u^{(k-1)} \right) \quad \text{with} \quad \mathcal{N}_v^* = \mathcal{N}_v \cup \{v\}$$

where $\sigma$ is an activation function and $W^{(k)}$ is a weight matrix for the $k^{\text{th}}$ iteration/layer.

After $k$ iterations the representation of a node captures the information contained in its $k$-hop neighborhood [e.g., see the illustration of a vertex rooted sub-tree in 5, Figure 1]. The node features at the output of the last iteration are aggregated into a graph-level representation using a *readout* function. To enforce permutation invariant hypotheses, it is common to employ the standard pooling functions as readouts — sum, mean, or max. In the next section, we consider possible extensions that would allow for learning readout functions jointly with other parameters of graph neural networks.

## 2.1 Neural Readouts

Suppose that after completing a pre-specified number of neighborhood aggregation iterations, the resulting node features are collected into a matrix $H \in \mathbb{R}^{M \times D}$, where $M$ is the maximal number of nodes that a graph can have in the dataset and $D$ is the dimension of the output node embedding. For graphs with less than $M$ vertices the padded values in $H$ are set to zero. We also denote with a vector $h \in \mathbb{R}^{M \cdot D}$ the flattened (i.e., concatenated rows) version of the node feature matrix $H$.

**Set Transformer Readouts.** Recently, an attention-based neural architecture for learning on sets has been proposed in [10]. The main difference compared to the classical attention model proposed by Vaswani et al. [12] is the absence of positional encoding and dropout layers. The approach can be motivated by the desire to exploit dependencies between set items when learning permutation invariant hypotheses on that domain. More specifically, other approaches within the deep sets framework typically embed set items independently into a latent space and then generate a permutation invariant hypothesis by standard pooling operators (sum, max, or mean). As graphs can be seen as sets of nodes, we propose to exploit this architecture as a readout function in graph neural networks. For the sake of brevity, classical attention models are described in Appendix D and here we summarize the adaptation to sets. The set transformers take as input matrices with items/nodes as rows and generate graph representations by composing attention-based encoder and decoder modules:

$$\text{ST}(H) = \frac{1}{K} \sum_{k=1}^{K} \left[ \text{DECODER} \left( \text{ENCODER} \left( H \right) \right) \right]_k \tag{2}$$

where $[\cdot]_k$ refers to computation specific to head $k$. The encoder-decoder modules are given by [10]:

$\text{ENCODER}(H) \coloneqq \text{MAB}^n(H, H) \quad \text{and} \quad \text{DECODER}(Z) \coloneqq \text{FF}\left( \text{MAB}^m \left( \text{PMA}(Z), \text{PMA}(Z) \right) \right)$

$\text{where} \quad \text{PMA}(Z) \coloneqq \text{MAB}(s, \text{FF}(Z)) \quad \text{and} \quad \text{MAB}(X, Y) \coloneqq A + \text{FF}(A)$

$\text{with} \quad A \coloneqq X + \text{MULTI-HEAD}(X, Y, Y) \,.$

Here, $H$ denotes the node features after neighborhood aggregation and $Z$ is the encoder output. The encoder is a chain of $n$ classical multi-head attention blocks (MAB) without positional encoding and dropouts. The decoder component employs a seed vector $s$ within a multi-head attention block to create an initial readout vector that is further processed via a chain of $m$ self-attention modules and a feedforward projection block (FF).

**Janossy Readouts.** Janossy pooling was proposed in [11] with the goal of providing means for learning flexible permutation invariant hypotheses that in their core employ classical neural architectures such as recurrent and/or convolutional neural networks. The main idea is to process each permutation of set elements with such an architecture and then average the resulting latent representations. Additionally, one could also add a further block of feedforward or recurrent layers to process the permutation invariant latent embedding of a set. Motivated by this pooling function initially designed for node aggregation, we design a readout that is approximately permutation invariant. More specifically, we consider MLP and GRU as base architectures and sample $p$ permutations of graph nodes. The Janossy readout then averages the latent representations of permuted graphs as follows:

$$\text{JANOSSY-MLP}(H) := \frac{1}{p}\sum_{i=1}^{p}\text{MLP}(h_{\pi_i}) \quad \text{or} \quad \text{JANOSSY-GRU}(H) := \frac{1}{p}\sum_{i=1}^{p}\text{GRU}(H_{\pi_i}), \quad (3)$$

where $\pi_i$ is a permutation of graph nodes and $h_{\pi_i}$ is the permuted and then flattened matrix $H$.

**Plain Feedforward/Recurrent Readouts.** Having proposed (approximate) permutation invariant readouts, we consider standard feedforward and recurrent neural architectures as well. Our MLP neural readout consists of a two-layer fully connected neural network (i.e., multi-layer perceptron) applied to the flattened node feature matrix $H$ denoted with $h$:

$$\text{MLP}(H) := \text{RELU}\big(\text{BN}_2(W_2 z_1 + b_2)\big) \quad \text{with} \quad z_1 = \text{RELU}\big(\text{BN}_1(W_1 h + b_1)\big) \quad (4)$$

where $W_1 \in \mathbb{R}^{(M \cdot D) \times d_1}$, $b_1 \in \mathbb{R}^{d_1}$, $z_1$ is the output of the first layer, $W_2 \in \mathbb{R}^{d_1 \times d_{\text{out}}}$, $b_2 \in \mathbb{R}^{d_{\text{out}}}$, $d_1$ and $d_{\text{out}}$ are hyperparameters, $\text{BN}_i$ is a batch normalization layer, and RELU is the rectified linear unit. In our experiments, we also apply Bernoulli dropout with rate $p = 0.4$ as the last operation within MLP. The GRU neural readout is composed of a single-layer, unidirectional gated recurrent unit (GRU, [13]), taking sequences with shape $(D, M)$ (i.e, the last two dimensions are permuted). We accept the input order on graph nodes as the order within the sequence that is passed to a GRU module. This recurrent module outputs a sequence with shape $(D, D)$, as well as a tensor of hidden states. The graph-level representation created by this readout is given by the last element of the output sequence.

In contrast to typical set-based neural architectures that process individual items in isolation (e.g., deep sets), the presented adaptive readouts account for interactions between all the node representations generated by the neighborhood aggregation scheme. In addition, the dimension of the graph-level representation can now be disentangled from the node output dimension and the aggregation scheme.

## 3  Experiments

We perform a series of experiments[1] to evaluate the effectiveness of the adaptive and differentiable neural readouts presented in Section 2.1 relative to the standard pooling functions (i.e., sum, max, mean) used for the aggregation of node features into a graph-level representation. In our experiments, we rely on two performance metrics: $R^2$ for regression tasks and Matthews correlation coefficient (MCC) for classifications tasks. As outlined in prior work [14, 15], these metrics can be better at quantifying performance improvements than typical measures of effectiveness such as mean absolute/squared error, accuracy, and $F_1$ score. To showcase the potential for learning effective representation models over graph structured data, we use in excess of $40$ datasets originating from different domains such as quantum mechanics, biophysics, bioinformatics, computer vision, social networks, synthetic graphs, and function call graphs. We focus on quantifying the difference in performance between the readouts relative to various factors such as: *i)* most frequently used neighborhood aggregations schemes, *ii)* the number of neighborhood aggregation iterations that correspond to layers in graph neural networks, *iii)* convergence rates measured in the number of epochs required for training an effective model, *iv)* different graph characteristics and domains from which the structured data originates, and *v)* the parameter budget employed by each of the neural

---

[1]The source code is available at https://github.com/davidbuterez/gnn-neural-readouts.

**Figure 1:** The performance of the best neural relative to the best standard readout on a collection of representative datasets from different domains. We use the ratio between the effectiveness scores ($R^2$ for QM9 and Matthew correlation coefficient otherwise), computed by averaging over five random splits of the data.

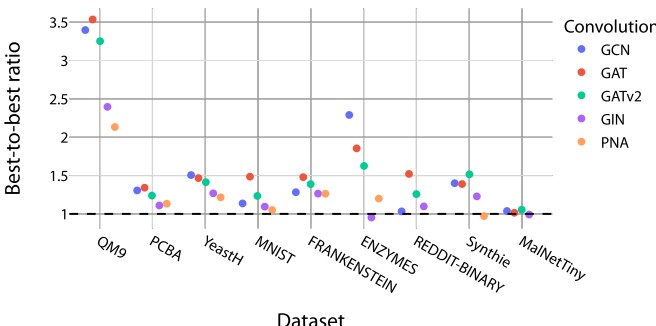

readouts. Moreover, we also evaluate a variety of neighborhood aggregation and readout schemes using large scale proprietary datasets with millions of compounds collected by primary screening assays. This is one of the first studies of a kind that demonstrates the behavior of standard graph neural networks on large scale datasets and illustrates some of the potential shortcomings that might be obfuscated by the small scale benchmarks that are typically used for evaluating their effectiveness in drug design tasks. In all of our experiments, we have used a node output dimension of $50$, which gives rise to a graph embedding dimension of that size for the sum, mean, and max readouts. For the adaptive readouts, the dimensionality of the graph representation is a hyperparameter of the model, which was fixed to $64$. For the sake of brevity, we present only a part of our analysis in this section and refer to the appendix for more detailed description of experiments and additional plots.

**Neural vs standard readouts across different datasets and neighborhood aggregation schemes**

The goal of this experiment is to evaluate the effectiveness of neural readouts relative to different datasets (39 in total, Appendix A), graph characteristics, and neighborhood aggregation schemes or convolutional operators. To this end, we opt for graph neural networks with two layers and compare neural to standard readouts across different graph convolutional operators: GCN [1], GAT [3], GATV2 [16], GIN [5], and PNA [17]. A detailed experimental setup (including loss functions, reporting metrics, and other details relevant for reproducibility) has been provided in Appendices B and F.

Figure 1 summarizes the result of this experiment over 9 representative datasets (please see Appendix G, Figures 1 to 3 for the results on the remaining 30 datasets, including other metrics). The figure depicts the ratio between the best neural and best standard readouts for each evaluated configuration (i.e., pair of dataset, convolution). We observe a considerable uplift in the performance on the majority of datasets. More specifically, in regression tasks we measure the performance using the $R^2$ score and on 36 configurations out of the possible 45 (i.e., $80\%$ of time) there is an improvement (with ratio $> 1$) as a result of employing neural readouts. In datasets where standard readouts fare better than the neural ones, the relative degradation is minor (i.e., less than $5\%$). In classification tasks, we use the MCC to measure the performance of graph neural networks and again observe an improvement as a result of employing neural readouts on 93 out of 147 (dataset, convolution) configurations ($\approx 63\%$ of time). We also observe that on 54 configurations where standard readouts are more effective than the neural ones that the relative degradation is below $10\%$ relative. We note that three models also failed to complete due to memory issues when using PNA, leading to a total of 147 configurations for the classification tasks (Appendix T). The minimum observed ratio between neural and standard readouts was $0.79$.

It is also worth noting that neural readouts come with hyperparameters that were not tuned/cross-validated in our experiments. For example, set transformer readouts can be configured by specifying the number of attention heads and latent/hidden dimensions. We have, throughout our experiments, followed standard practices and selected such hyperparameters to be powers of two, tailored to the dataset size (more details can be found in Appendix F, Table 4). This was also, in part, motivated by previous knowledge from bio-affinity prediction tasks with graph neural networks. Thus, it is likely that the performance can be further improved by hyperparameter tuning. We also emphasize that for this experiment the architecture is kept fixed across datasets (number of graph layers, hidden

**Figure 2:** The panel on the left illustrates the parameter budget of GNNs that does not account for the readouts, while varying the layer type and depth (QM9 dataset). The number of parameters for the MLP readout is represented using dashed lines parallel to the $x$-axis (slightly higher when using PNA as the output node dimension must be divisible by the tower hyperparameter). The panel on the right compares the performance of graph neural networks with set transformers (ST) and plain MLPs as readouts, while varying the number of parameters in the readout layer. This illustrative example has been obtained using the ENZYMES dataset and demonstrates that the effectiveness is not aligned with the number of trainable parameters in the readout layer but the type of architecture realizing it. For example, the ST 1−MAB model employs a single MAB-block encoder and decoder, being simpler in terms of the number of parameters than ST COMPLEX and more complex than ST MINIMAL (Appendix C). The MLP configurations are reported using the format MLP ($d_1$, $d_{out}$) (Appendix F).

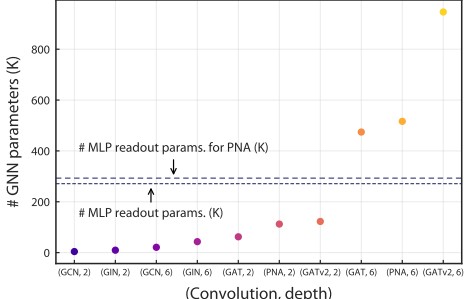

| Readout | # Params. | Avg. MCC |
|---|---|---|
| ST MINIMAL | 365K | 0.35 |
| ST 1−MAB | 628K | 0.34 |
| ST COMPLEX | 1154K | 0.36 |
| MLP (32, 32) | 130K | 0.41 |
| MLP (64, 32) | 260K | 0.44 |
| MLP (128, 64) | 525K | 0.40 |
| MLP (64, 128) | 267K | 0.47 |

dimensions, etc.), while only varying the graph layer type and readout layer. In addition, over-smoothing is a known problem for vanilla graph neural networks. To avoid any hidden contribution from over-smoothing correction techniques, we opted for shallower architectures. We also note that two-layer graph neural networks performed well on tasks such as bio-affinity prediction on the 1+ million scale datasets (i.e., can be sufficiently expressive). To validate that additional expressiveness due to more layers can be successfully exploited by adaptive readouts, we performed a separate suite of experiments where we varied the depth (see the next experiment/section). We conclude the discussion of this experiment with an insight into the trainable parameter budgets for a selection of neural readouts. For MLP, with the exception of GCN and GIN which use an extremely small number of trainable weights, the parameter budget/count is on the same scale as the rest of the graph neural networks (see the left panel in Figure 2). Furthermore, simply increasing the number of parameters does not necessarily improve the performance (see the right panel in Figure 2, ST vs MLP rows).

**Neural vs standard readouts relative to the number of neighborhood aggregations iterations**

The goal of this experiment is two-fold: *i*) assessment of the effectiveness of neural readouts relative to the depth of graph neural networks, and *ii*) validation of the observations in the experiments with two layer graph neural networks, i.e., the improvements characteristic of neural readouts are not due to underfitting that can be caused by employing a small number of neighborhood aggregation iterations along with standard readouts. We perform the experiment with various graph convolutional operators using datasets with different number of instances, ranging from 600 to 132,480 graphs on ENZYMES and QM9, respectively. In these experiments, the only variable graph neural network hyperparameter is the number of neighborhood aggregation iterations (i.e,. depth). We have also performed this experiment on one of the proprietary bio-affinity datasets with 1.5 million graphs (see Appendix N, Figure 19). Figure 3 summarizes the results for the QM9 experiment (see Appendix H, Figure 4 for ENZYMES). The trends seen for standard readouts are mirrored for the neural ones as the number of neighborhood iterations increases, i.e., deeper graph neural networks can lead to more expressive models, irrespective of the readout type.

**Convergence of training algorithms for graph neural networks relative to readouts**

As outlined in Section 1, it might be challenging to learn a permutation invariant hypothesis over graphs using simple and non-adaptive readout functions such as sum, mean, or max. Here, we argue that such functions might be creating a tight link between the resulting graph-level representations and the computed node features, in the sense that: *i*) it takes a long time for the graph-level representation to adjust to the prediction task and *ii*) it is difficult for the graph-level representation to diverge from the learned node features and adapt to the target property. To validate this hypothesis, we recorded the graph representations for a random molecule from the QM9 dataset in each training epoch (for multiple convolutions and readouts). We computed the Euclidean distances between the initial graph

**Figure 3:** Increasing the number of neighborhood aggregation iterations or convolutional layers has different effects on QM9. The trends observed for the standard readouts are mirrored for the neural ones, particularly for the most powerful one on this dataset (SET TRANSFORMER). For GCN, GAT, and GATV2, the performance improves as the depth is increased to 6 layers and drops afterwards. GIN is generally stable relative to the number of layers, while PNA has an initial performance improvement (up to 3, 4 layers) and then plateaus.

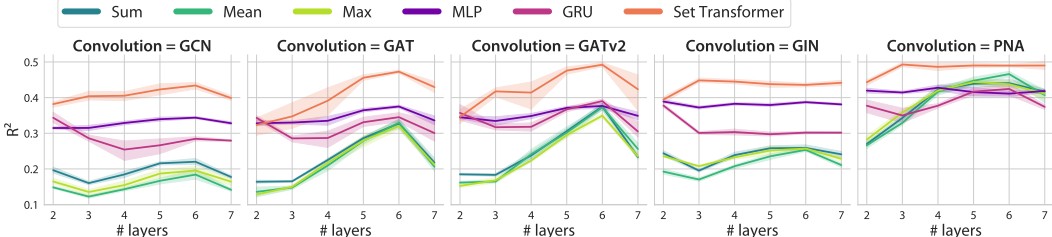

**Table 1:** $R^2$ (mean $\pm$ standard deviation) on QM9, using the ST MINIMAL and ST COMPLEX architectures.

| Aggregator | # Heads | GCN | GAT | GATv2 | GIN | PNA |
|---|---|---|---|---|---|---|
| ST MINIMAL | 1 | $0.20 \pm 0.02$ | $0.17 \pm 0.01$ | $0.20 \pm 0.01$ | $0.22 \pm 0.01$ | $0.27 \pm 0.01$ |
| | 4 | $0.27 \pm 0.01$ | $0.23 \pm 0.02$ | $0.27 \pm 0.02$ | $0.32 \pm 0.01$ | $0.38 \pm 0.02$ |
| | 8 | $0.35 \pm 0.01$ | $0.24 \pm 0.04$ | $0.29 \pm 0.01$ | $0.35 \pm 0.01$ | $0.43 \pm 0.01$ |
| | 12 | $0.37 \pm 0.01$ | $0.26 \pm 0.03$ | $0.27 \pm 0.03$ | $0.37 \pm 0.01$ | $0.43 \pm 0.01$ |
| ST COMPLEX | 1 | $0.18 \pm 0.02$ | $0.16 \pm 0.02$ | $0.20 \pm 0.01$ | $0.24 \pm 0.01$ | $0.28 \pm 0.01$ |
| | 4 | $0.35 \pm 0.01$ | $0.27 \pm 0.02$ | $0.29 \pm 0.02$ | $0.35 \pm 0.00$ | $0.42 \pm 0.01$ |
| | 8 | $0.38 \pm 0.01$ | $0.30 \pm 0.04$ | $0.32 \pm 0.02$ | $0.39 \pm 0.01$ | $0.44 \pm 0.01$ |
| | 12 | $0.37 \pm 0.02$ | $0.32 \pm 0.02$ | $0.33 \pm 0.02$ | $0.40 \pm 0.01$ | $0.45 \pm 0.01$ |

embedding (i.e., the first epoch) and all subsequent epochs (Appendix K, Figure 11), as well as between consecutive epochs (Appendix K, Figure 12). Our empirical results indicate that GNNs with standard readouts take hundreds of epochs to converge (500 to $1,000$), with minor changes in the graph representation from one epoch to another. In contrast to this, the models employing neural readouts converge quickly, typically in under 100 epochs. Moreover, the learned representations can span a larger volume, as shown by the initial and converged representations, which are separated by distances that are orders of magnitude larger than for the standard readouts.

**The importance of (approximate) permutation invariance in adaptive readouts**

The goal of this experiment is to obtain an insight into the effects of adaptive readouts that give rise to permutation invariant hypothesis spaces on the network's ability to learn a target concept. To this end, we exploit the modularity of SET TRANSFORMERS and consider architectures with 1 to 12 attention heads, as well as a different number of attention blocks: an ST MINIMAL model with one MAB in the encoder and no MABs in the decoder, and ST COMPLEX with two MABs in both the encoder and decoder (for 2-layer GNNs). Table 1 provides a summary of the results for this type of readouts on the QM9 dataset (detailed results can be found in the appendix). With a small number of attention heads (1 and 4), all models with SET TRANSFORMER readouts are able to outperform the ones with standard pooling functions. However, over different convolutions the models with few attention heads are outperformed by the ones with MLP and GRU readouts that do not enforce permutation invariance. Increasing the number of heads to 8 or 12 leads to the best performance on this dataset for all graph convolutions. However, the relative improvement gained by increasing the number of attention heads beyond 4 is generally minor, as is the uplift gained by adding a self-attention block. We also evaluated the impact of enforcing approximate permutation invariance by Janossy readouts on the QM9 dataset (Appendix T, Table 27). The Janossy variants presented in Section 2.1 outperform the three standard readouts in both mean absolute error and $R^2$, but they score lower than the other neural readouts.

**Robustness to node permutations for non-permutation invariant readouts**

The objective of this experiment is to evaluate the stability of readouts that either do not enforce permutation invariance of the hypothesis space (i.e., MLP and GRU readouts) or do so only approximately (i.e., JANOSSY readouts). To this end, we generate 50 random node permutations (relative to the order induced by the graphs generated from the canonical SMILES) for 50 randomly selected molecules from the QM9 dataset and run these through the previously trained models with two

**Figure 4:** A summary of the error distributions for predictions made on random permutations of 50 randomly selected molecules from the QM9 dataset. The error is computed as the absolute difference between the predicted and target labels. The models are fully trained, two-layer graph neural networks. Due to permutation invariance, the predictions made for a given molecule are identical for sum/mean/max, regardless of the permutation. The variance for these readouts is a result of the differences in the predicted labels for the considered 50 molecules. Panels **(a)** and **(b)** reflect different strategies of generating node permutations.

**(a)** For each molecule, we generate 50 different graphs using random permutations of the nodes originating from the canonical SMILES representation.

**(b)** For each molecule, we generate graphs corresponding to different non-canonical SMILES. The number varies from 20 to $1,000$ (per-molecule).

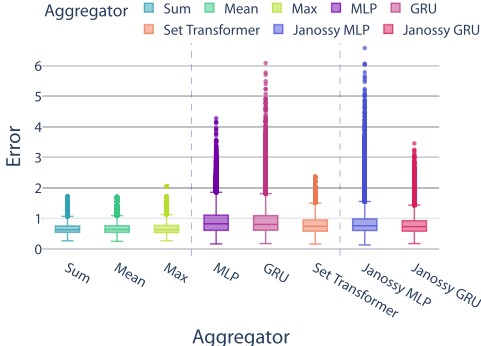

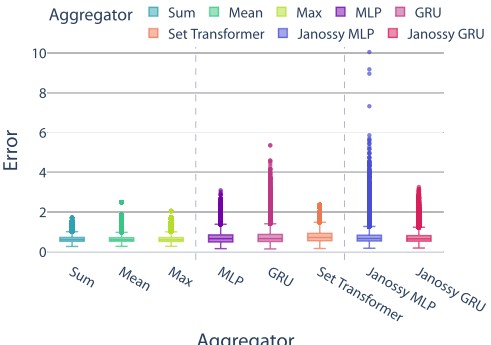

layers (see above for details, neural vs standard readouts). As all graph molecular representations originate from a canonical SMILES string (simplified molecular-input line-entry system, [18]), we also generate permuted graphs by using non-canonical SMILES. For the latter, we have applied repeated sampling of permuted SMILES using RDKit until the total number converged, resulting in permuted molecular graphs. Depending on the molecule, it is possible to generate as little as 20 different representations or as many as $1,000$ (all the resulting graphs are used in the analysis). Permutations due to non-canonical SMILES represent a subset of arbitrary permutations that are specific to the chemistry domain. For each molecule and random permutation, we computed the error as the absolute difference between the predicted and target labels. Figures 4a and 4b summarize the results of this experiment and show that the plain readouts (i.e., MLP and GRU) can be negatively affected by some random permutations. Interestingly, the error for MLP is greatly accentuated on certain QM9 prediction tasks, such as U0, U298, H298, and G298, while the error is very similar to sum, mean, and max on tasks such as CV, ZPVE, and R2 (see Appendix J, Figure 9 for more details). The Janossy readouts are trained not only on the original fixed-order graphs but also on 25 random node permutations computed during training (i.e., a different stage compared to the evaluation presented here). We observe that the JANOSSY GRU readout improves upon the plain GRU, leading to a distribution that is more similar to the SET TRANSFORMER readout, with a reduced number of outliers. In contrast, the JANOSSY-MLP readout appears to be performing the worst in terms of robustness to node permutations. A breakdown of results over different neighborhood aggregation schemes and convolutions can be found in Appendix J, Figures 6 and 7. In summary, all neural readouts exhibit significantly reduced errors for the permutations originating from randomly generated non-canonical SMILES compared to the arbitrary node permutations, despite being trained only using the canonical representations. Moreover, we encounter many different (permuted) molecular representations that attain the minimal error values for MLP readouts (see Appendix J, Figure 8).

**The effectiveness of graph neural networks and readouts on multi-million graph datasets**
In this experiment, we extend our empirical analysis with three AstraZeneca high-throughput screening assays against varied protein targets. Each dataset consists of 1 to 2 million molecular measurements, such that each molecule is associated with a single label (scalar) corresponding to its activity value (see Appendix A, Table 1 for details). We trained both non-variational and guided variational graph autoencoder [19, 20] models (denoted by GNN and VGAE, respectively) for 200 epochs on the entirety of the available data for each dataset, with the goal of assessing the ability of graph neural networks to learn expressive graph representations at this scale (Figure 5, with the other 5 models in Appendix M). Our analysis considers the performance relative to different neighborhood aggregation schemes or convolutional operators as well as the total number of such iterations that correspond to depth in graph neural networks (separately, see Appendix N, Figure 19). Figure 5 summarizes the results of this experiment and indicates that the models using standard non-adaptive readouts (sum,

**Figure 5:** Train loss for the VGAE models trained on a proprietary dataset with $\approx 1$ million molecular graphs.

mean, or max) generally struggle to model molecular data at the $1+$ million scale, as reflected in the plateaued training losses for each of the models. Depending on the dataset and the neighborhood aggregation scheme, the MLP readout significantly outperforms the other readout functions, with a train loss that rapidly decreases in the first 50 epochs. The SET TRANSFORMER readout is the second-best choice, converging with a slightly a slower rate and occasionally diverging. The GRU readouts offer only a slight improvement compared to the standard readout functions. When training both variational and non-variational models with deeper architectures (2, 3, and 4 layers, excluding $\mu$- and $\sigma$-layers for the VGAE), we do not observe significant benefits introduced by additional iterations of neighborhood aggregation schemes. This is in line with the study that introduced these datasets [20], which also evaluated multiple convolutional operators and numbers of such iterations/layers. Instead, as indicated previously, the largest benefits are generally associated with more powerful neighborhood aggregation schemes. The results are further supported by the training metrics (MAE, $R^2$) after 200 epochs (see also Appendix O, Tables 8 to 13), which provide an insight into the ability of graph neural networks to fit signals on $1+$ million scale datasets. For example, our results for the VGAE GCN model trained on the proprietary dataset with $\approx 1$ million graphs show that neural readouts lead to an improvement in the $R^2$ score from 0.33 to 0.78, with $\approx 1.5$ million graphs from 0.07 to 0.64, and with $\approx 2$ million graphs from 0.06 to 0.52.

## 4 Discussion

We have presented an extensive evaluation of graph neural networks with adaptive and differentiable readout functions and various neighborhood aggregation schemes. Our empirical results demonstrate that the proposed readouts can be beneficial on a wide variety of domains with graph structured data, as well as different data scales and graph characteristics. Overall, we observed improvements in over two thirds of the evaluated configurations (given by pairs consisting of datasets and neighborhood aggregation schemes), while performing competitively on the remaining ones. Moreover, we have empirically captured and quantified different aspects and trade-offs specific to adaptive readouts. For instance, the effectiveness of adaptive readouts that do not enforce permutation invariance of hypothesis spaces (MLP and GRU) indicates that it might be possible to relax this constraint for certain tasks. A primary candidate for relaxation are molecular tasks, which are also one of the most popular application domains for graph neural networks. Molecules are typically presented in the canonical form, a strategy also adopted by popular open-source frameworks such as RDKit. Thus, the graph that corresponds to any given molecule comes with a fixed vertex ordering when generated from the canonical SMILES. Our analysis suggests that neural readouts trained on canonical representations can learn chemical motifs that are applicable even to non-canonical inputs, or in other words, generally applicable chemical knowledge. It should be noted here that the canonical representations differ greatly even for extremely similar molecules (see also Appendix R), such that it is improbable that the graph neural networks are learning simple associations based on position or presence of certain atoms. Instead, it might be the case that the networks can learn certain higher-level chemical patterns that are strictly relevant to the task at hand.

We have also discussed possible domain-specific interpretations for the effectiveness of models with adaptive readouts on some tasks. For instance, certain molecular properties tend to be approximated well by neural readouts, while others remain more amenable to standard pooling functions such as sum. Chemically, properties such as the internal energy, enthalpy, or free energy are generally considered additive (e.g., can be approximated by a sum of pairwise bond energies) and extensive

(increasing almost linearly with the number of atoms). Such properties are a good fit for standard readouts. Other properties, such as the highest occupied molecular orbital and lowest unoccupied molecular orbital (HOMO and LUMO, respectively) tend to be localized and are considered non-additive, such that a single atom can potentially completely alter the property, or not influence it at all. Popular problems such as bio-affinity prediction are also regarded as highly non-linear. Overall, this interplay suggests hybrid readouts for future research, where certain properties would be learned by functions such as sum, while others are left to more flexible neural readouts.

Regarding practical details such as the choice of the most suitable adaptive readout function for a given dataset, our empirical results indicate that larger and more complex (relative to the number of nodes per graph and dimension of node features) regression tasks see more pronounced performance improvements with adaptive readouts (based on statistically significant results from linear regression models detailed in Appendix Q). We were, however, unable to observe a similar pattern for the considered classification tasks. Thanks to its potential for composing highly expressive neural architectures, SET TRANSFORMER is likely better suited for larger datasets. However, graph neural networks with that readout function tend to occasionally experience divergence on very large datasets or deep architectures (6+ layers), which can most likely be fully resolved with parameter tuning, especially the latent/hidden dimension of the attention mechanism. An avenue that might be promising for further study is pre-training readout networks, such that they can be quickly deployed on related tasks and fine-tuned. One starting point could be pre-training on large molecular databases, such as subsets of GDB-17 [21] with inexpensive to compute molecular tasks (generated with RDKit, for example) as prediction targets, or unsupervised variations.

When it comes to related approaches, the majority of recent efforts have been focused on neighborhood aggregation schemes. This step also requires permutation invariance and it is interesting that a related work by Hamilton et al. [22] has considered relaxation to that constraint and employed an LSTM neural network to aggregate neighborhoods and produce node features. Along these lines, Murphy et al. [11] introduced Janossy pooling, a permutation-invariant pooling technique for neighborhood aggregation, designed for node classification tasks. Perhaps the most related to our direction and readouts is the concurrently developed work by Baek et al. [23] on graph multi-set transformers, i.e., a multi-head attention model based on a global pooling layer that models the relationship between nodes by exploiting their structural dependencies. For the purpose of measuring and fixing the over-squashing problem in graph neural networks, Alon and Yahav [24] proposed a fully-adjacent layer (each pair of nodes is connected by an edge), which greatly improved performance and resembles our use of the MLP readout. Prior work has also considered tunable $\ell_p$ pooling [25] and more restrictive universal readouts based on deep sets [26]. However, neither of these approaches offers a comprehensive empirical evaluation at the scale provided here.

As adaptive readouts introduce a new differentiable component to graph neural networks, future studies might focus on analyzing properties such as transferability and interpretability. In our empirical study, we did not consider such experiments due to conceptual and practical differences. Conceptually, one of the main motivating factors for studying transferability is a scenario where the graph (network) size changes over time. This is typically encountered in recommendation systems or knowledge graphs which are not considered in our paper. Regarding the graph-to-graph transferability, there are domain-specific particularities that need to be considered. For example, learning on small graphs and transferring to larger graphs is not often required in chemical tasks, as most chemical regression benchmarks and real-world applications use only very small organic molecules (e.g., $< 30$ atoms or nodes for QM9). There is also the requirement of selectivity, where an active molecule should bind only to a selected target and possible issues can arise with transferring a notion of similarity over the space of molecules that encodes activity to a completely different target. Moreover, there have been reports where the impact of learning (with graph neural networks) certain transferable chemical substructures (scaffolds) was not beneficial [27]. Practically, transferability has been most often studied with node-level tasks [28], while here we focus on graph-level predictions. Overall, we believe that studying the influence of adaptive readouts on transferability is interesting for future studies. Regarding the interpretability, we have in this work focused on allowing for more flexibility in neural readouts (Appendix K) and the structuring effect on the learned latent space, possibly making it amenable to clustering and other downstream tasks (Appendix L).

**Acknowledgments**: We are grateful for access to the AstraZeneca Scientific Computing Platform. David Buterez has been supported by a fully funded PhD grant from AstraZeneca.

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
