# OpenReview forum: "Graph Neural Networks with Adaptive Readouts"
_NeurIPS.cc/2022/Conference — NeurIPS 2022 Accept_

### Official Review · Reviewer_hKnu · 2022-07-10

**Rating:** 6
**Confidence:** 4
**Soundness:** 3 good
**Presentation:** 4 excellent
**Contribution:** 2 fair

**Summary:**

This paper provides a comprehensive empirical study of graph neural networks (GNN) with adaptive readouts. Graph neural networks traditionally use non-adaptive readouts based on simple functions (like sum, max, mean) to preserve permutation invariance. Experiments in this paper demonstrate that permutation invariance can be traded with performance improvement by using learnable, adaptive readout functions based on neural architectures.

**Questions:**

1. A brief description of Matthew correlation coefficient would improve readability.
2. The bio-affinity prediction task is not clear to me and I believe some details are missing. For instance, what is SMILES representation in this context? I suggest adding appropriate details or references to explain this task.
3. Can the authors comment on the relevance of neighborhood aggregations with a multilayer perceptron based DENSE readout layer? In this case, I am concerned that neighborhood aggregations might become irrelevant with the learning performance primarily attributable to the DENSE layer for at least some of the experiments.
4. I recommend adding discussions on transferability of GNNs with adaptive readouts.

**Limitations:**

Yes.

**Strengths And Weaknesses:**

Strengths:
This paper combines traditional GNN architectures with three adaptive readouts in its empirical evaluations and therefore, there are no significant technical advancements. The main strength of the paper is the comprehensive empirical analysis for datasets that span multiple domains (e.g. bioinformatics, computer vision, quantum mechanics etc.) from the existing literature. The experiments demonstrate considerable improvements in performance using adaptive readouts over that for standard readout functions. The experiments are well explained and the accompanying discussions are illuminating from an applied machine learning perspective.


Weaknesses:
1. I think the property of transferability of GNNs (see ref. [a] below) merits significant discussion in the paper. Adopting adaptive readouts in GNNs affects transferability in conjunction with permutation invariance.
2. The focus of the paper is on improvement in performance using adaptive readouts. However, in some applications, the interpretability of the results can be equally or more important than performance. An example of this scenario is brain age prediction, where the error in brain age may have prognostic value and potentially predict health outcomes in disease (see ref. [b] below).


[a] Ruiz, Luana, Luiz Chamon, and Alejandro Ribeiro. "Graphon neural networks and the transferability of graph neural networks." Advances in Neural Information Processing Systems 33 (2020): 1702-1712.

[b] Lee, J., Burkett, B.J., Min, HK. et al. Deep learning-based brain age prediction in normal aging and dementia. Nat Aging 2, 412–424 (2022). https://doi.org/10.1038/s43587-022-00219-7

---

> ### Author Response · Authors · 2022-08-02
> **Response to Reviewer 3 (part 1)**
>
> ### Transferability & Question 4
> Transferability with performance guarantees is certainly an interesting and relevant property of GNNs. In our current evaluation, we did not consider such experiments due to conceptual and practical differences. Conceptually, one of the main motivating factors for studying transferability is a scenario where the graph (network) size changes over time. This is typically encountered in recommender systems or knowledge graphs which are not considered in our paper. Regarding the graph-to-graph transferability, there are domain-specific particularities that need to be considered. For example, learning on small graphs and transferring to larger graphs is not often required in chemical tasks, as most chemical regression benchmarks and real-world applications use only very small organic molecules (for example, <30 atoms or nodes for QM9). There is also the requirement of selectivity, where an active molecule should bind only to a selected target and possible issues can arise with transferring a notion of similarity over the space of molecules that encodes activity to a completely different target. Moreover, there have been reports where the impact of learning (with GNNs) certain transferable chemical substructures (scaffolds) was not beneficial [**]. Practically, transferability has been most often studied with node-level tasks, such as in ref. [a] indicated by the reviewer, while here we focus on graph-level predictions. Overall, we believe that studying the influence of adaptive readouts on transferability is interesting for future studies, and we will ensure that it is discussed in our updated manuscript.
>
> [**] Sakai, M., Nagayasu, K., Shibui, N. et al. Prediction of pharmacological activities from chemical structures with graph convolutional neural networks. Sci Rep 11, 525 (2021).
>
> ### Interpretability & Weakness 2
> We considered that establishing the performance uplifts of the proposed methods is one of the main priorities and thus we dedicated a large amount of space to this endeavour. At the same time, we appreciate the importance of interpretability, especially for sensitive tasks such as drug bio-affinity prediction. So far, we have taken some first steps towards analysing the interpretability. Firstly, we examined the learnt graph representations of adaptive readouts (Appendix L) and concluded that they have the potential to be more expressive, as they explore a larger hypothesis space.
> Secondly, we visualised the graph representations using dimensionality reduction (UMAP) in 3 dimensions on public bio-affinity molecular datasets (Appendix M), which suggested that the adaptive readouts have a structuring effect on the graph latent space, possibly making it more amenable to clustering and other downstream tasks. Future studies might focus on exploring different aspects of the adaptive readouts. For example, as neural networks, they can be subjected to an analysis of the learnt weights or attention scores.
>
> ### Question 1 (MCC)
> Please see [*] and the response to Reviewer 1 (Question 1) for more details on MCC and reasons for opting for this performance metric.
>
> [*] D Chicco and G Jurman (BMC Genomics, 2020). The advantages of the Matthews correlation coefficient (MCC) over F1 score and accuracy in binary classification evaluation.
>
> ### Question 2 (SMILES)
> SMILES strings are a compact way of representing a molecule and an alternative input format to a graph. For more details, please see [***]. In our paper, we use them to extract the graph connectivity and atom information (all discussed models are GNNs using graph representations). We will include references and better description of the bio-affinity related tasks considered here (e.g., internal energy, enthalpy, highest and lowest occupied molecular orbital, binding affinity relative to protein targets, etc.).
>
> [***] Weininger D (1988). "SMILES, a chemical language and information system. 1. Introduction to methodology and encoding rules". Journal of Chemical Information and Computer Sciences. 28 (1): 31–6.

---

> > ### Author Response · Authors · 2022-08-02
> > **Response to Reviewer 3 (part 2)**
> >
> > ### Question 3 (MLP vs neighbourhood aggregations)
> > The scale of the y-axis in Figure 2 fails to fully illustrate the improvement achieved with MLP readouts as a result of increasing the number of neighbourhood aggregations (the readout hyperparameters are kept fixed). As an example, the uplifts (R-squared) for the MLP readout and GAT layers are as follows: 2 layers = 0.328, 3 layers = 0.330, 4 layers = 0.335, 5 layers = 0.365, 6 layers = 0.375.
> >
> > While the improvement is modest, the experiment shows that this type of readout function can benefit from more iterations of neighbourhood aggregation. Moreover, the MLP builds a representation while accounting for all node representations and as the experiment suggests it might be possible to learn a good graph representation with fewer iterations of neighbourhood aggregations (please also see Appendix L for a related discussion on expressivity). We have also discussed some chemical properties that might be more amenable to MLP readouts and where insisting on permutation invariance via standard pooling operators might be sub-optimal (see Section 4, lines 326-350).
> >
> > Please also find more details on the parameter budget and adaptive readouts in the response to Reviewer 1 (Question 2).

---

### Official Review · Reviewer_rDRe · 2022-07-11

**Rating:** 6
**Confidence:** 3
**Soundness:** 3 good
**Presentation:** 2 fair
**Contribution:** 4 excellent

**Summary:**

The paper introduces novel adaptive and differentiable readout functions for Graph Convolutional Networks that rely on neural networks and which do not necessarily result in permutation invariant hypothesis space.
The authors provide an extensive empirical evaluation on 40 datasets which involves some standard readout/pooling policies and different neighborhood aggregation schemes.


**Questions:**

It is not clear why some pooling techniques like the one reported in [19], [20[ [21], and [22] were not considered in the empirical evaluation.

In section 3 (rows 200-201) the authors state “Thus, it200 is likely that the performance can be further improved by hyperparameter tuning.”, therefore I wonder whether (and how) the authors performed a validation phase and how the hyper parameters used in the various experiments were selected.

In the subsection “Neural vs standard readouts across different datasets and neighborhood aggregation schemes'' the authors use two-layer GNN. This choice clearly introduces a bias in the valuation strategy. How much does it impact the obtained results?

**Limitations:**

The paper turns out to be very interesting and I appreciate the considerable amount of comparisons and evaluation that the authors performed to assess the proposed approach. The point is that all the reported analyses and evaluations require more than a 9-page conference paper to be described and discussed. Indeed the authors report a considerable quantity of the results (crucial to understanding the proposed discussion) and a part of the description of the architecture in the appendix. In my opinion that makes the paper difficult to read. The organization of the information and its division between the main paper and appendix have to be revised.

Some references were missing, e.g the wide use of graph-level aggregators like DiffPool (Ying, R., You, J., Morris, C., Ren, X., Hamilton, W.L., Leskovec, J.: Hierarchical graph representation learning with differentiable pooling (2019)) and the Sortpool (Zhang, M., Cui, Z., Neumann, M., Chen, Y.: An end-to-end deep learning architecture for graph classification. In: Thirty-Second AAAI Conference on Artificial Intelligence (2018))  that have to be at least discussed. Same for the recently proposed SOM-based aggregator (Pasa, L., Navarin, N., Sperduti, A.: Som-based aggregation for graph convolutional neural networks. Neural Computing and Applications pp. 1–20 (2020)).

**Strengths And Weaknesses:**

Strengths:
- Several datasets and different domains considered in the experiments
- The authors empirically analyze several different aspects, including an ablation study that highlights the impact of various parts of the model.

Weaknesses:
- Crucial parts of the results are reported only in the appendix
- Some aspects of the empirical evaluation have to be clarified (e.g validation strategy)
- Missing some references
- Some common graph pooling methods are not considered in the experimental comparison

---

> ### Author Response · Authors · 2022-08-02
> **Response to Reviewer 2 (part 1)**
>
> ### Question 1 (pooling techniques from [19-22])
> When designing the experiments we were faced with a limited compute budget and have opted to proceed with a representative method from each of the following readout classes: i) non-adaptive (sum, mean, max), ii) adaptive with permutation invariance (Set Transformer being the most representative), iii) adaptive and approximately permutation invariant (Janossy pooling), and iv) feedforward and recurrent neural networks that do not necessarily give rise to permutation invariant hypothesis spaces. This choice allowed us to carry out an extensive evaluation in terms of different benchmark datasets, graph convolution types, number of neighbourhood aggregation iterations, as well as other experiments (for example, Appendix L and M). We think that these baselines are sufficient to demonstrate the potential of adaptive readouts across different domains and also raise an interesting question on the utility of standard feedforward and recurrent networks as readouts.
>
> ### Question 2 (hyperparameter tuning and validation strategy)
> The comment on possible improvements with further hyperparameter tuning (lines 200-201) refers only to adaptive readouts. Figure 1 shows that GNNs with such readouts are, even without hyperparameter tuning, significantly more effective than identical graph convolutional architectures with non-adaptive readouts (sum, mean, max). A detailed description of the experimental design can be found in Appendices B and F, covering train/validation/test splits along with other reproducibility details. We have, in general, followed well adopted practices and selected hyperparameters of neural readouts to be powers of two (more details can be found in Appendix Table 4), tailored to dataset size. This was also, in part, motivated by previous knowledge from bio-affinity prediction tasks with GNNs.
>
> ### Question 3 (bias in the validation strategy)
> We have performed experiments with fixed (e.g., see Figure 1, two layers) and varying number of graph convolutions (e.g., see Figure 2). The reviewer's comment probably refers to experiments in Figure 1 but these are not to be taken independently of the ones reported in Figure 2 (see also Appendix O and Appendix Figure 4), where we varied the depth of GNNs while keeping the readouts fixed. In both sets of experiments, our empirical results indicate the benefits of employing adaptive readouts. For some graph convolution types (e.g., PNA) and benchmark datasets (e.g., QM9) the performance of MLP and GRU readouts is on par with non-adaptive ones (see Figure 2), but even in those cases Set Transformers are significantly more effective than non-adaptive readouts.  In addition to this, over-smoothing is a known problem for vanilla GNNs. To avoid any hidden contribution from over-smoothing-correction techniques, we preferred shallower GNNs. Finally, we note that two-layer GNNs performed well on tasks such as bio-affinity prediction on the 1+ million scale datasets (i.e., can be expressive). In general, one can observe that the performance curves of adaptive readouts mirror the trends of the non-adaptive ones, but with considerably higher scores, indicating that the additional GNN expressivity due to more layers can be successfully exploited by the adaptive readouts.
>
> ### Limitations (main part vs Appendix)
> When it comes to the split of the content between the main part and Appendix, we should have an extra page for the camera-ready version and will ensure that some illustrations from the Appendix are moved to the main paper and that all the results there are properly referenced. We agree with the reviewer that the scale of the empirical contribution might be fitting to a journal paper but would also like to point that the two other reviewers are satisfied with a summary of the results provided in the main paper (at least judging by their presentation scores).

---

> > ### Author Response · Authors · 2022-08-02
> > **Response to Reviewer 2 (part 2)**
> >
> > ### Limitations (SortPool, DiffPool, etc.)
> > We thank the reviewer for the additional references that will be included into the next version of the paper. Tables 3 and 4 below (from this response; results are split into two tables to increase readability, following the format presented in the paper) summarise the results of our experiment with SortPool that was done during the rebuttal period, with two-layer GNNs on QM9. Our results indicate that the approach is not competitive with the adaptive readouts considered here. For some of the other approaches, it was challenging to secure bug-free code or an implementation compatible with the PyTorch Geometric variation of the graph convolutional layers (e.g., DiffPool requires a dense adjacency matrix which is incompatible with the considered GNN layers).
> >
> > **Table 3.** SortPool results for GCN, GAT, and GATv2.
> > \begin{array}{|c|c|c|c|c|c|c|}
> >     \hline
> >     & \text{GCN} & \text{GCN} & \text{GAT} & \text{GAT} & \text{GATv2} & \text{GATv2} \\\\ \hline
> >     \text{Agg.} & \text{MAE} & \text{R}^2 & \text{MAE} & \text{R}^2 & \text{MAE}  & \text{R}^2 \\\\ \hline
> >         \text{SortPool} & 0.75 \pm 0.01 & 0.07 \pm 0.02 & 0.74 \pm 0.01 & 0.09 \pm 0.02 & 0.73 \pm 0.01 & 0.12 \pm 0.03  \\\\ \hline
> > \end{array}
> >
> > **Table 4.** SortPool results for GIN and PNA.
> > \begin{array}{|c|c|c|c|}
> >     \hline
> >     & \text{GIN} & \text{GIN} & \text{PNA} & \text{PNA} \\\\ \hline
> >     \text{Agg.} & \text{MAE}  & \text{R}^2  & \text{MAE} & \text{R}^2\\\\ \hline
> >         \text{SortPool} & 0.72 \pm 0.01 & 0.13 \pm 0.02 & 0.69 \pm 0.01 & 0.19 \pm 0.02 \\\\ \hline
> > \end{array}

---

### Official Review · Reviewer_GVMn · 2022-07-22

**Rating:** 8
**Confidence:** 5
**Soundness:** 3 good
**Presentation:** 3 good
**Contribution:** 3 good

**Summary:**

(**Disclaimer**: I was assigned to this paper as an emergency reviewer;
sorry for being slightly more terse than I would like to be ideally)

This paper presents an empirical analysis of the predictive performance
of graph neural networks when switching to different readout functions.
A set of such readout functions is investigated, some of which do *not*
satisfy permutation invariance by design. Surprisingly, the performance
of such readout functions turns out to be beneficial on some graph data
sets and some learning tasks!


**Questions:**

1. Is there a reason why MCC was used instead of accuracy differences or
   accuracy ratios?

2. To what extent could the observed performance increases by explained
   by having more parameters available for a model to fit? I am
   mentioning this because it appears that one recommendation is to use
   a set transformer, i.e. a permutation-invariant readout function.
   This slightly changes the message of the paper to 'More parameters
   for readout functions are better,' whereas initially, the goal was to
   see whether permutation invariance is actually needed. Please comment
   on this.

**Limitations:**

Limitations are not directly addressed. A brief statement on the broader
impact of the work would be welcome. At the same time, I do not foresee
any adverse outcomes that arise specifically because of this work.

**Strengths And Weaknesses:**

The main strength of the paper lies in the strong experimental section,
which challenges the status quo. The use of pooling functions that satisfy
permutation invariance has been deeply ingrained in the graph learning
literature. Thus, an investigation of their efficacy in practice is
highly appreciated and useful.

The main weakness of the paper is a missing discussion about parameter
budgets; to a certain extent, some of the observed differences could
potentially also be explained by having 'more expressive' readout
functions (permutation invariance notwithstanding). This should be
clarified but can easily be accomplished within the conference reviewing
cycles (see questions below).

Moreover, a more specific discussion of 'takeaway messages' could be
added, making some recommendations for practitioners in light of the
observed phenomena.

In summary, I believe that this paper requires only minor modifications
before being ready for publication. It will make an excellent
contribution to the conference.

## Detailed Comments

- When discussing graph expressivity, I would suggest to rather state
  that certain certain GNNs are *not more expressive than the WL test*,
  i.e. a 'shallow' method without trainable weights. Moreover, the test
  is not as computationally efficient as GNNs; this should be rectified.

- The beginning of Section 2 could potentially be condensed to make
  more space for experiments or discussions, which are critical for this
  type of paper.

- The fact that differences are statistically significant (Appendix R)
  should be mentioned more prominently in the main text in Figure 1.

- Figure 1 could potentially also be improved by showing standard
  deviations of the 'best-to-best ratio'.

- Figure 2 (and other figures as well) should use a consistent colour
  scheme and ensure that the *new* readout functions are highlighted
  somewhat prominently.

- More information about the dimensionality of output vectors should be
  provided, in particular when discussing aspects like convergence or
  Euclidean distance (see also next point).

- For the calculation of Euclidean distances (Appendix I), more details
  about the models and the dimensionalities should be provided. It is
  possible (though not likely) that the changes in distances can also be
  explained by some 'nuisance' dimensions in the outputs. Moreover, this
  type of analysis would lend itself well to a repetition in order to
  get an idea of the variance in the data.

## Minor Comments

- Eq. (3) could be improved; there are several terms to keep track of at
  the moment; $\mathrm{SAB}$ could for instance just be substituted by
  $\mathrm{MAB}$.

- Space permitting, a delineation to the results from the Janossy
  pooling paper could be provided. Having read both papers, I recall
  that Janossy pooling *also* discusses the merits of a different
  readout function.

- Why are some $p$-values zero in the appendix? Is this a problem of
  typesetting the table?

- Please check the bibliography again; there are some citations that
  should refer to published work but point to a preprint at the moment
  (for example, Alon and Yahav [20] is now published at ICLR; the work
  by Kipf and Welling [1] was likewise published at ICLR).

- When discussing WL, it might be interesting to cite a [recent survey
  that summarises the connections of WL to machine learning](https://arxiv.org/abs/2112.09992).

---

> ### Author Response · Authors · 2022-08-02
> **Response to Reviewer 1 (part 1)**
>
> ### Strengths and weaknesses & Question 2 (parameter budgets)
> Table 1 (from this response) summarizes the parameter budget for the experiments with Dense (MLP) readouts, reported in Figure 2 (see also Appendix O and Appendix Figure 4). We will include this table into the next version of the paper and extend the existing discussion of this experiment. Employing six graph convolutional layers translates to a 5 to 10-fold increase in the number of GNN parameters (this does not refer to readout parameters) compared to two layers, depending on the type of convolution. Moreover, six-layer graph neural networks with GAT, GATv2, and PNA convolutions employ (approximately) at least double the number of parameters assigned to the MLP readout. The results in Figure 2 indicate that the MLP readout coupled with only two graph convolutional layers performs on par with a graph neural network with six layers and standard non-adaptive readouts. Moreover, the latter neural network (e.g., see GAT, GATv2, and PNA) has approximately double the number of parameters compared to the former.
>
> **Table 1.** D = output dimension of GNN. The number of parameters for the MLP readout is computed according to the formula: M * D * H1 + H1 * H2, given without an additive bias for brevity, where additionally M = maximum number of nodes in the graph, H1 = output dimension of the first MLP layer, and H2 = output dimension of the second MLP layer. PNA uses a slightly altered D to satisfy the layer's internal assertions when using 5 towers. Num. = Number
>
> \begin{array}{|c|c|c|c|c|c|}
> \hline
> \text{Layer type} & \text{Num. layers} & \text{Num. GNN params (K)} & \text{Num. MLP readout params (K)} & \text{D}  \\\\ \hline
> \text{GCN}        & 2         & 4.1                   & 271                               & 32 \\\\ \hline
> \text{GCN}        & 6         & 20.9                  & 271                               & 32 \\\\ \hline
> \text{GIN}        & 2         & 9.5                   & 271                               & 32 \\\\ \hline
> \text{GIN}        & 6         & 43.1                  & 271                               & 32 \\\\ \hline
> \text{GAT}        & 2         & 62.2                  & 271                               & 32 \\\\ \hline
> \text{GAT}        & 6         & 474.2                 & 271                               & 32 \\\\ \hline
> \text{PNA}        & 2         & 112.3                 & 293                               & 35 \\\\ \hline
> \text{PNA}        & 6         & 516.3                 & 293                               & 35 \\\\ \hline
> \text{GATv2}      & 2         & 122.5                 & 271                               & 32 \\\\ \hline
> \text{GATv2}      & 6         & 946.5                 & 271                               & 32 \\\\ \hline
> \end{array}
>
> In Appendix J (Table 17), we have also performed experiments with Set Transformers with different numbers of attention heads and self-attention blocks. This set of experiments demonstrates that already with a single attention head and a single self-attention block, graph neural networks with adaptive readouts can outperform the ones with standard pooling functions -- sum, mean, and max.
>
> In Appendix Figure 4, as well as other examples such as the BACE dataset in Appendix Table 6 and the two datasets in Appendix Table 10 (this list is non-exhaustive), we observed that the MLP readout outperformed the Set Transformer readout. During the rebuttal period, we performed another experiment on the ENZYMES dataset, using GNNs with two GCN layers and the parameter budget described in Table 2 (from this response), presented alongside the obtained performance metric (MCC). Although the Set Transformer used a considerably larger parameter budget (approximately x4.5), this did not lead to increased performance over the MLP. We will include and extend this analysis in the next version of the paper.
>
> We are also happy to include additional experiments that would help clarify this concern and would appreciate if the reviewer could recommend some that we could run in the background.
>
> **Table 2.**
>
> \begin{array}{|c|c|c|}
> \hline
> \text{Readout}        & \text{Num. readout parameters} & \text{Average MCC} \\\\ \hline
> \text{MLP}            & 260\text{K}                  & 0.44        \\\\ \hline
> \text{Set Transformer} & 1.2\text{M}                  & 0.375      \\\\ \hline
> \end{array}
>
> ### Question 1 (MCC)
> We selected this metric based on [*], which identified the MCC as a more informative classification metric than the accuracy or the $F_1$ score and a reasonable default choice. This is currently detailed in Appendix B, but we will attempt, space allowing, to clarify this in the main text (there should be an extra page available for the camera-ready version).
>
> [*] D Chicco and G Jurman (BMC Genomics, 2020). The advantages of the Matthews correlation coefficient (MCC) over F1 score and accuracy in binary classification evaluation.

---

> > ### Author Response · Authors · 2022-08-02
> > **Response to Reviewer 1 (part 2)**
> >
> > ### Strengths and weaknesses (takeaway messages)
> > We are happy to extend our current discussion that already offers some guidelines for practitioners (e.g., see lines 326-350), especially for chemically-oriented tasks that appear to be benefiting the most from adaptive readouts. We would greatly appreciate further clarification on the takeaway messages that the reviewer thinks would be beneficial for practitioners and will ensure they are part of the next revision.
> >
> > ### Detailed and minor comments
> > We thank the reviewer for the detailed comments that will improve the quality of our manuscript. We will ensure these are incorporated into the camera-ready version of the manuscript.
> >
> > As for the WL test, we will resolve this ambiguity in the next version. The computational efficiency argument does not apply to GNNs but to the isomorphism test that can decide a large number of isomorphism classes.
> >
> > In the current submission, the dimension of the GNN outputs is set to 32. We will provide more details on the experimental setup in the camera-ready version and add an illustration depicting the variance.
> >
> > As for the p-value question and reference to the appendix (see minor comments), the value of 0 means that the observed value was below machine precision (will clarify this). We will revise our bibliography, update the listed references, and include the suggested survey on the WL algorithm.

---

> > > ### Comment · Reviewer_GVMn · 2022-08-05
> > > **Thanks**
> > >
> > > Thanks for the thorough answers! I appreciate them and can state that this helped in clarifying matters for me.
> > >
> > > Concerning your question on the 'takeaway' messages: your paper challenges the status quo by showing that readout functions do not necessarily (!) have to be `sum` or `max`. It would now be interesting if you could make actionable recommendations here, for instance "For molecular graphs, we recommend X". This, I think, would help in connecting the paper and demonstrate that it establishes new community standards.

---

> > > > ### Comment · Reviewer_GVMn · 2022-08-09
> > > > **Updated score**
> > > >
> > > > Forget to mention this in the previous comment, and I am not sure whether you get a notification for this: in light of the rebuttal, I am happy to raise my score. If you could briefly comment on the 'takeaway messages' part, I'd appreciate it.

---

> > > > > ### Author Response · Authors · 2022-08-09
> > > > > **Thanks and takeaway messages**
> > > > >
> > > > > We thank the reviewer for the clarification on "takeaway messages". We will be expanding the existing discussion on readout functions in the camera-ready version. While our experiments are detailed, they do not cover all the possible factors that might influence a decision on the readout type. We will, however, endeavour to make recommendations and hypotheses taking into account all of our experiments. It is very encouraging that our empirical results support the use of adaptive readouts across different domains (e.g., set transformers are a convenient choice that retains permutation invariance and should be at the top of the list of considered readouts). Other aspects relevant to problems involving molecular graphs are the specifics of the target property being modelled. As outlined in our discussion, some properties are influenced by a part of the molecule and standard readouts might not be the best choice for such problems (in fact, the MLP can be a better adaptive choice than the set transformer readout). The selected course of action should, thus, be informed by the target property and task type (regression or classification), dataset size, graph structures characteristic of the problem, etc. We will ensure that our recommendations are supported with extensive empirical results and examples while covering as many of the factors as possible.

---

### Author Response · Authors · 2022-08-02
**Official response**

We would like to thank all the reviewers for their insightful comments and suggestions, which will help us to improve the quality of the paper and its presentation. We have addressed the individual concerns raised by the reviewers in our detailed response below.

---

### Meta-Review · Area_Chair_YaEt · 2022-08-31

**Recommendation:** Accept
**Confidence:** Certain

**Metareview:**

The paper proposes the use of an adaptive readout function in GNNs together with extensive empirical work to support it. The reviewers all found the paper interesting and are generally in favor of accepting it (with one marking strong accept with high confidence). Therefore, I recommend the paper be accepted, and encourage the authors to take into account the reviewer comments (as they have also indicated in their responses) when preparing the camera ready version. In particular, I would like to encourage them to use the extra page given there to reconsider the split of materials between main paper and appendix.

**Award:**

No

---

### Decision · Program_Chairs · 2022-09-14

Accept